# Toward a consistent modeling framework to assess multi-sectoral climate impacts

Erwan Monier [1], Sergey Paltsev [1], Andrei Sokolov[1], Y.-H. Henry Chen[1], Xiang Gao[1], Qudsia Ejaz[1], Evan Couzo[1,4], C. Adam Schlosser[1], Stephanie Dutkiewicz[1], Charles Fant[1], Jeffery Scott[1], David Kicklighter[2], Jennifer Morris[1], Henry Jacoby[1], Ronald Prinn[1] & Martin Haigh[3]

Efforts to estimate the physical and economic impacts of future climate change face substantial challenges. To enrich the currently popular approaches to impact analysis—which involve evaluation of a damage function or multi-model comparisons based on a limited number of standardized scenarios—we propose integrating a geospatially resolved physical representation of impacts into a coupled human-Earth system modeling framework. Large internationally coordinated exercises cannot easily respond to new policy targets and the implementation of standard scenarios across models, institutions and research communities can yield inconsistent estimates. Here, we argue for a shift toward the use of a self-consistent integrated modeling framework to assess climate impacts, and discuss ways the integrated assessment modeling community can move in this direction. We then demonstrate the capabilities of such a modeling framework by conducting a multi-sectoral assessment of climate impacts under a range of consistent and integrated economic and climate scenarios that are responsive to new policies and business expectations.

[1] Joint Program on the Science and Policy of Global Change, Massachusetts Institute of Technology, 77 Massachusetts Ave, Cambridge, MA 02139, USA. [2] The Ecosystems Center, Marine Biological Laboratory, 7 MBL St, Woods Hole, MA 02543, USA. [3] Shell International, Shell Centre, York Road, London, SE1 7NA, UK. [4] Present address: University of North Carolina, Asheville, One University Heights, Asheville, NC, 28804, USA. Correspondence and requests for materials should be addressed to E.M. (email: emonier@mit.edu) or to S.P. (email: paltsev@mit.edu)

Estimating the impacts of climate change is challenging because they span a large number of economic sectors and ecosystems services, and can vary strongly by region[1–3]. Many integrated assessment models (IAMs) rely on simple box climate models and use a damage function approach to estimate a social cost of carbon[4] that relates changes in emissions to economic damage[5]. These models are useful. For example, the US EPA and other government agencies use these estimates to evaluate the climate benefits of rulemakings[2]. But this approach has also attracted criticism[6,7] as the existing literature offers sparse theoretical support and provides scant empirical evidence for a specification of economic damages, especially at temperatures outside the historical range.

Another widely used approach relies on model inter-comparison projects (MIPs) that apply the results of detailed biogeophysical models. They can offer valuable insights into specific climate impacts (e.g., the Agricultural Model Inter-comparison and Improvement Project (AgMIP) for agriculture[8]). However, these exercises suffer from a rigid and complex framework, driven by the need for international coordination, so they must rely on a limited number of socio-economic scenarios, like the four representative concentration pathways (RCP) scenarios[9]. Since the developers and the users of these scenarios come from different research groups and disciplinary communities, major inconsistencies in their implementation, such as socio-economic assumptions and ecosystem characteristics, can easily occur. For example, when a common land-use scenario is implemented in different Earth system models (ESMs), differences appear in cropland and pastureland areas because of the different interpretations of land-use classes by the ESMs[10]. The resulting differences in the carbon cycle and land-use forcing are thus difficult to interpret. Also, each of the four RCP scenarios was developed by a different IAM group, and their projections of future air pollutant emissions are inconsistent with one another[11], making comparisons of air quality among RCP scenarios of little value. Since many climate impact assessments do rely on MIPs, and are not done within a cohesive IAM framework, these inconsistencies can contaminate analysis of the benefits of climate policies.

Furthermore, the MIPs lack flexibility, and responsiveness to changes in economic and environmental policies (like the recent Paris Agreement), and thus they are of limited usefulness in analysis of policy choice. In addition, because of their single sector focus these exercises do not capture important inter-dependencies, linkages and feedbacks, and this lack of integration among sectors is likely to lead to misrepresentation of climate impacts[12]. Moreover, IAMs that use a single-sector macro-economic representation of the global economy lack the cap-ability for evaluation of particular sectors of the economy where damages occur. Finally, there is little effort and limited capability to synthesize the many MIPs into an overarching assessment of climate impacts across sectors of the economy, which further limits the information value these exercises bring to the decision process.

In recent years, major efforts have been pursued toward the development of consistent modeling frameworks to assess climate impacts using a new generation of IAM, which place a greater emphasis on representing the coupled human-Earth system (CHES) model—essentially IAM version 2.0. Such modeling frameworks include both a detailed representation of economic activities, to track inter-sectoral and inter-regional links, and a detailed representation of the various physical, chemical, and biological components of the Earth system that are impacted by human activity. The aim is to provide a tight integration among three communities that, though internally collaborative, have remained largely isolated from one another: the IAM, the Earth System Modeling (ESM), and the impacts, adaptation, and vul-nerability (IAV) communities. An advantage of such an approach is that research groups can construct new scenarios of climate change and conduct climate impact assessments, while ensuring consistent treatment of interactions among population growth, economic development, energy and land system changes and physical climate impacts. Such new scenarios can provide improved estimates of the impact of current and proposed international agreements, and other aspects of climate policy[13].

To provide an example of such a CHES modeling framework and demonstrate its capabilities, we examine socio-economic and climate change impacts under a range of consistent and inte-grated economic and climate scenarios using the MIT Integrated Global System Model (IGSM)[14–17]. The IGSM couples a human system model to an ESM of intermediate complexity (EMIC), and links to a series of geospatially resolved physical impact models (see Methods section). While we showcase the IGSM, other models could be used as well, as other IAM groups have made similar improvements in the integration of the coupled human and Earth systems. Examples include the Integrated Model to Assess the Greenhouse Effect (IMAGE)[18], the Global Change Assessment Model (GCAM)[19], the model for energy supply strategy alternatives and their general environmental impact (MESSAGE)[20] and the Asia Pacific Integrated Model (AIM)[21]. In this paper, we first discuss strategies for coupling between human and ESMs and the improved integration of geospatially resolved physical impact models. We then present a multi-sectoral climate impact assessment focusing on ocean acidification, air quality, water resources and agriculture under consistent and integrated economic and climate scenarios that are responsive to new policies and business expectations. This example then provides a basis for arguing the advantages of such a shift toward a consistent CHES modeling framework to assess climate impacts.

## Results

**Strategies for coupling human and ESMs**. The human system component of a CHES model should represent the world's economy, disaggregated into multiple regions and with sectoral detail (i.e., agriculture, services, industrial and household trans-portation, energy-intensive industry). It also should include trade, investments, savings, and consumption decisions, as well as abatement of greenhouse gases (GHGs) through the imple-mentation of policies like carbon taxes, emissions trading, mea-sures to support specific technologies (e.g., wind, solar, carbon capture), and regional fuel and emissions standards. The Earth system component should simulate the coupled atmosphere, ocean, land (including rivers and lakes) and cryosphere (sea ice, land ice, permafrost), including the dynamical and physical processes (i.e., river flow, ocean eddies, cloud processes, erosion), chemical processes (chemical gases and aerosols), biogeochemical processes (life-mediated carbon-nutrient dynamics), and bio-geophysical processes (life-mediated water and energy balance).

In practice, because state-of-the-art ESMs are computationally expensive, a CHES model can be built by coupling a human system model to a simplified model of the climate system and to specific impact models for key ecosystems and sectors of the economy (Fig. 1). Different coupling strategies exist[22], from off-line one-way information exchange between research commu-nities to fully coupled modeling approaches that yield more or less instantaneous (depending on the timestep of the coupling) two-way interactions between the human and Earth system components. Other strategies include improving the representa-tion of the Earth system in IAMs or improving the representation of societal elements within ESMs. Beyond the challenge of coupling the human and Earth systems, an important

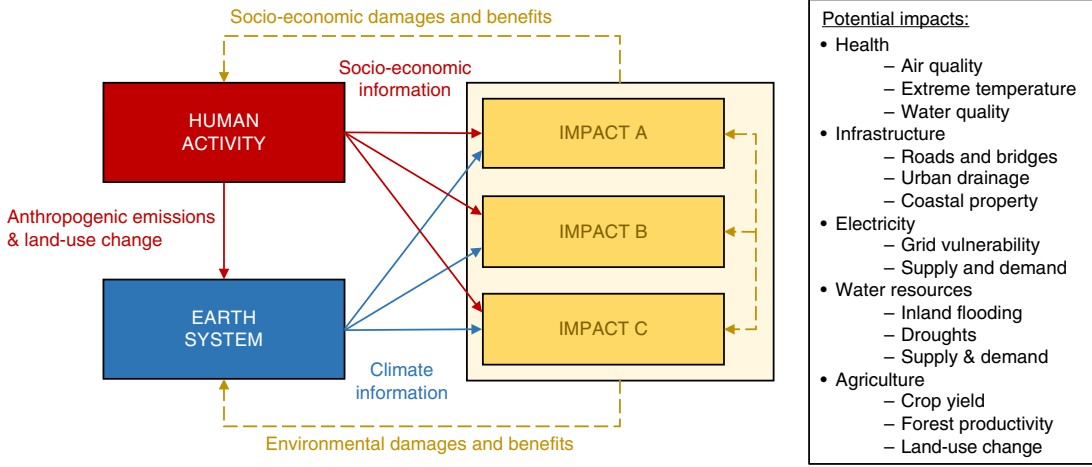

**Fig. 1** Conceptual representation of a coupled human-Earth system model with improved integration of physical impact models. Different coupling strategies between the various components of the modeling framework are represented by the different arrows. Potential impacts are listed in the right box

characteristic of CHES models should be a detailed representation of the biophysical impacts of climate change, spanning key economic sectors and ecosystem services. (Additional details on the various coupling strategies and their advantages are provided in the Supplementary Table 1.)

While the coupling strategy remains an important issue, and full coupling between the human and Earth systems is an aspirational goal, the effort will not be very insightful if it involves dubious damage functions, like it does in social cost of carbon models[23,24]. Also, full coupling raises many additional challenges, e.g., difficulties of coupling different software systems, complexities of representing the cascading of uncertainty among components of the system, and differences in temporal and spatial scale of the various components. As a result, full coupling is generally limited to a specific pathway, like the land system[25]. In addition, full integration is not warranted unless there is evidence that it would substantially change the estimates of climate impacts. In the process of developing a CHES model, therefore, a one-way coupling where physical impacts of climate change are explicitly modeled, but do feed back onto GHG emissions and the climate system (e.g., land-use change), is a useful first step. A salient response from the one-way testing will then warrant exploration of two-way coupling which, if found to produce significant new insights, can be incorporated in subsequent versions of the model. A similar approach is suggested to interactions among impact models, for example between air quality and agriculture.

**Improved integration of physical impact models.** To simulate regional changes in temperature and precipitation, the IGSM can be combined with statistical emulation techniques (pattern scaling) to represent the differences in the regional patterns of change exhibited by different climate models[26,27], or it can be coupled to a 3-dimensional atmospheric model when 3-dimensional and highly-resolved temporal climate information is required or to assess the role of natural climate variability at the regional scale[28]. To examine the fate of the oceans under future climate change, the IGSM includes a 3-dimensional dynamical, biological, and chemical ocean general circulation model capable of physically estimating global and regional changes in ocean acidification, the meridional overturning circulation, or the structure of phytoplankton communities[29–31].

To analyze the co-benefits of GHG mitigation on air quality, the IGSM is linked to a 3-dimensional atmospheric chemistry

model[32,33] that simulates, among others, changes in ground level fine particulate matter ($PM_{2.5}$) concentrations, where the human system model is combined with a detailed emissions inventory to provide anthropogenic emissions of precursors. Because the influence of climate change on air quality has been found to be small compared to the impact of reduction in emissions, we do not couple the atmospheric chemistry model to the climate model in the IGSM, instead using fixed meteorological fields for a chosen year (e.g., 2010) or set of years that capture distinct climate conditions (e.g., El Niño/La Niña/neutral year) for all simulations. To assess the changes in water resources driven by climate change and socio-economic drivers (e.g., population increase) the IGSM includes a river basin scale model of water resources management[34,35], representing the competition for water among industry, agriculture and domestic use in the face of changes in water demand and water supply in 282 Assessment Sub-Regions (ASRs) over the globe—but that can also run at a more spatially resolved capacity over specific regions[36,37].

Finally, to investigate the future of agriculture, the IGSM is coupled to a global gridded process-based terrestrial ecosystem/ biogeochemistry carbon-nitrogen model, which simulates the impact of climate (temperature, precipitation, and solar radiation), atmospheric chemistry ($CO_2$ fertilization and ozone damage) and nitrogen limitation on crop yield, and accounts for land-use change adaptation decisions made by the human system model[38,39]. Because ozone damage has been identified as a major stressor on land productivity[40], it is included in this analysis. However, the impact of land-use change on the climate system, through GHG emissions and changes in surface albedo[10,41], is not included because it has not been demonstrated to be a key feedback on agricultural productivity. (More details are provided in Methods section.)

**Integrated economic and climate scenarios.** The integrated economic and climate scenarios are developed following three typical approaches in business, government and academia to explore the future: the desired, a normative scenario aimed at limiting global warming in 2100 to 2 °C from pre-industrial (named 2C) using a global economy-wide carbon tax; the likely, an outlook based on existing policy, here an assessment of the results from the UN COP-21 meeting[42] (named Paris Forever), assuming no additional climate policy after 2030, resulting in 3.5 ° C warming in 2100, emphasizing that the current pledges are not sufficient to meet the goal to stay "well below 2 °C"[43]; and the

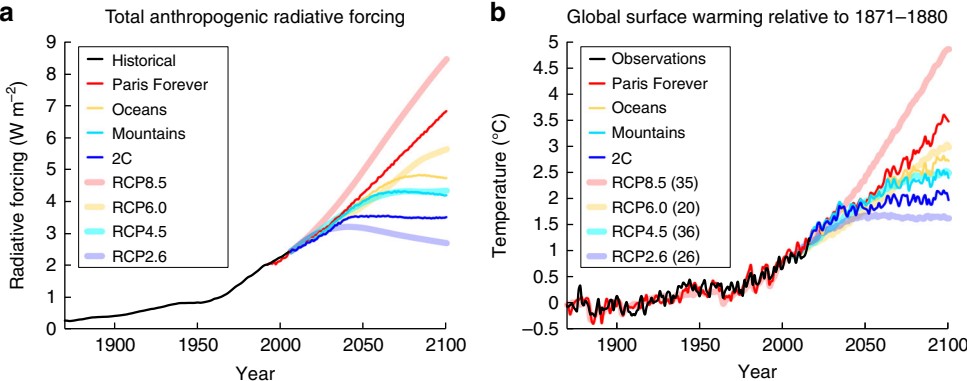

**Fig. 2** Radiative forcing and global temperature change. Time series of **a** total anthropogenic radiative forcing (W m$^{-2}$) and **b** global surface warming (°C) relative to 1871–1880 for the Paris Forever, Oceans, Mountains and 2C integrated economic and climate scenarios along with the 4 RCP scenarios. The multi-model ensemble mean is shown for global mean temperature in the 4 RCPs. Numbers in parentheses represent the number of climate models in the ensemble under each RCP scenario. Temperature observations are from the Berkeley Earth Surface Temperatures (BEST)[92]

plausible, two exploratory scenarios to assess the potential development of low-carbon energy technologies (named Oceans and Mountains)[44], with warmings of 2.7 °C and 2.4 °C in 2100, respectively. Compared to the RCP scenarios (Fig. 2), these scenarios are responsive to and grounded in the latest existing climate policies (UN COP-21). They do not include a business-as-usual scenario like the RCP8.5, and they avoid scenarios based on unproven negative emissions technologies, like the RCP2.6. We provide a few applications of integrated impact assessments focusing on ocean acidification, air quality, water resources and agriculture (Fig. 3). (Additional details on the scenarios are provided in Supplementary Table 2, and GHG emissions are shown in Supplementary Fig. 1.)

**Multi-sectoral climate impact assessment.** Under the Paris Forever scenario, the global ocean pH would drop to levels under 7.9 by 2100, which would significantly impact all calcareous phytoplankton that are the base of the ocean food chain, and would damage or destroy coral reefs[45], but the ocean acidification is significantly reduced under the 2C scenario. China and India, two countries that currently experience severely polluted ambient air (with annual mean concentrations of PM$_{2.5}$ greater than air quality standards over major areas), would see increased pollution by 2100 under the Paris Forever scenario, with PM$_{2.5}$ concentrations doubling in many regions. However, these countries would experience significant co-benefits of imposing a carbon tax under the 2C scenario, with reductions in co-emitted air pollutants including PM$_{2.5}$. By 2100, the population exposed to water stress is generally projected to increase by several hundred million under most scenarios, mainly driven by increases in water demand from a growing population. However, the use of different climate models—through statistical emulation techniques—results in contrasting estimates of the impact of climate mitigation, because of differences in regional patterns of precipitation change. Under a relatively dry climate model pattern (model N), the higher warming scenario is associated with stronger regional decreases in precipitation and thus increased water scarcity over densely-populated areas. Emissions mitigation reduces the degree of water scarcity. On the other hand, this finding is reversed under a relatively wet climate model pattern (model M), thus motivating the implementation of much larger ensemble simulations to properly assess these risks[46]. Finally, large increases in temperature, exceeding the damaging temperature thresholds for crop productivity[47,48], and major ozone damage[40] are projected under the Paris Forever scenario. Even under cropland relocation, extension and intensification, the overall global crop yield (over

crop land areas) decreases by 2100. Emissions mitigation results in substantial reductions in warming and surface ozone concentrations, so land-use change adaptation can lead to benefits to the agriculture sector. (Additional analyses for all scenarios are provided in Supplementary Figs. 2, 3 and 4, with a summary of the major findings in Supplementary Table 3.)

Our results show varying levels of agreement with existing impact assessments, especially those within the Inter-Sectoral Impact Model Intercomparison Project (ISIMIP) framework[49]. The ocean acidification analysis is consistent with existing ESM intercomparison under the RCP scenarios[50]. The population exposed to water stress is generally in agreement with the analysis from a large ensemble of global hydrological models forced by five global climate models under the RCP scenarios[51], but it lies on the lower end because we explicitly integrate the biogeophysical modeling of water resources with a water resources management model and thus optimize water resources. Also, our finding that there are conditions under which GHG mitigation could increase water scarcity resonates with an analysis focusing on the US[52]. The major co-benefits of reducing GHG emissions on air quality are consistent with existing estimates[11], although the actual magnitude of the co-benefits can vary substantially among studies because of differences in the scenarios and differences in the treatment of criteria pollutant emissions by different IAMs. Finally, the climate impacts on agricultural productivity differ from AgMIP analyses[53] because our estimates include ozone damage and land-use change adaptation. Few studies bring together estimates of climate impacts across ecosystems and sectors of the economy under a consistent modeling framework, using consistent socio-economic and climate scenarios.

## Discussion
Assessing climate change impacts is a challenging task, and many researchers are cautious about reducing impacts to a monetary value. Despite being an active area of research, there is little theory to guide the damage functions needed to directly translate change in global mean temperature to impacts on gross domestic product (GDP), and in many cases arbitrary functional forms and corresponding parameter values are chosen[54]. In contrast, we focus on understanding the chain of actual physical changes at the regional and sectoral levels and then estimating the economic impacts, thus bridging the gaps among the IAM, ESM, and IAV communities. Our results show that the projected climate impacts vary dramatically across the globe, with large uncertainties in the physical climate impacts associated with differences in the

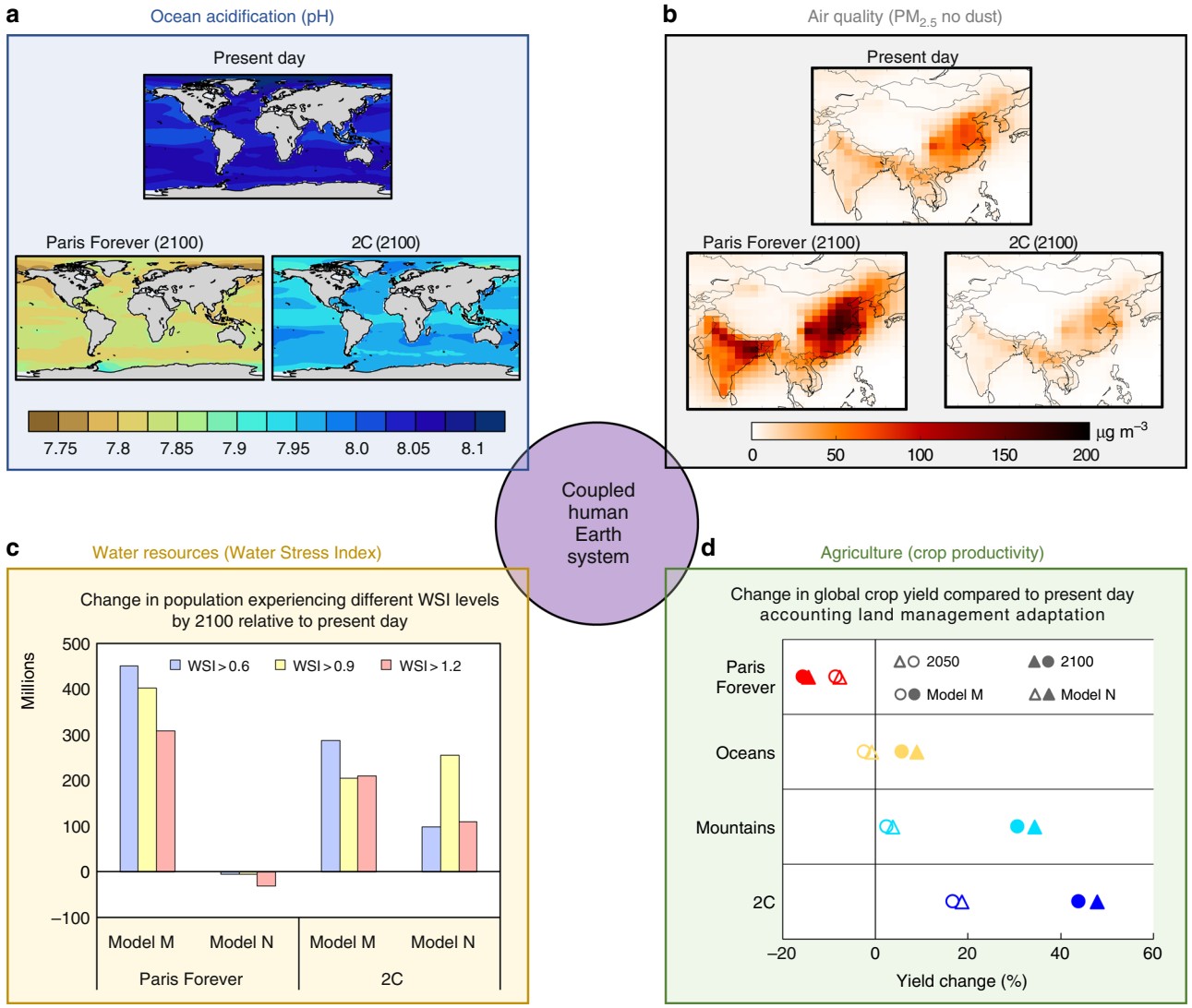

**Fig. 3** Sample multi-sectoral climate impact assessment using the improved coupled human-Earth system modeling framework. Climate impacts are focusing on **a** ocean acidification (pH), **b** air quality over China and India (PM$_{2.5}$, no dust), **c** water resources (Water Stress Index) and **d** agriculture (crop productivity). Except for the air quality analysis, present day, 2050 and 2100 correspond to, respectively, 10-year averages over the 2001–2010, 2046–2055 and 2091–2100 periods. For the water resources and agriculture climate impact assessments, results are shown for a dry (model M) and wet (model N) climate model via statistical emulation techniques

magnitude and patterns of climate change from different climate models, thus putting in question the adequacy of damage functions based on global mean temperature. These results also support the need to rely on probabilistic ensembles of climate simulations to determine the full range of outcomes and move into quantitative climate risk assessments. Such probabilistic impact assessment has been conducted with the IGSM for specific regions of the world[46]. Furthermore, our analysis shows that more stringent emission reduction scenarios (Oceans, Mountains, 2C) are successful in mitigating a large portion of these climate impacts. These projections demonstrate the relative value of each emissions mitigation policy by relying on consistent economic and climate projections that provide a sound physical basis for the estimates of climate impacts. Finally, they do not require internationally coordinated modeling efforts that can be cumbersome and time consuming, and that sometimes lag the implementation of climate policies in the real world.

This strategy to move toward a more integrated and self-consistent representation of the coupled human and Earth systems, with a geospatially resolved physical representation of

climate impacts, has largely emerged from a recent research focus on the food-energy-water (FEW) nexus and outcomes from MIPs, like ISIMIP, which have shown the importance of linking IAMs with physical impact models[53,55–57]. The improvement of existing IAMs, implementing the paradigm shift that a CHES approach represents, is ongoing in many integrated assessment modeling research groups[58], with different levels of integration, number of impacts considered, and speed of model development[59–64]. Thus far, however, these efforts have focused largely on individual sectors of the economy, like energy for heating and cooling[65–67], water resources[68–71] or air pollution[72].

While an uncertainty analysis is beyond the scope of this paper, we recognize that the climate impact results discussed above are subject to substantial uncertainty. The common approach to address uncertainty in climate impact studies is through multiple impact model ensembles driven by multiple climate model ensembles (e.g., ISIMIP/AGMIP). Because of the lack of flexibility and responsiveness of these coordinated multi-model exercises, we argue that a complementary approach is to use model emulators (e.g., crop yield emulators[73] or climate emulators[26,27,46])

along with large model ensembles—perturbing the physics, parameters, and initial conditions—within a consistent CHES modeling framework[13,27,74,75]. Such an approach could not only help quantify parametric or scenario uncertainty, but also address structural uncertainty (associated with the use of different models) by using emulators to reproduce and account for the varying behavior of different models.

We further argue that it is possible to develop computationally-efficient models, which represent the various essential components of the Earth system and provide a physical representation of climate impacts, albeit in reduced forms (e.g., EMIC or emulators). These modeling frameworks can be used for risk analysis instead of relying on box models and dubious damage functions. At the same time, computationally demanding process-based impact models are still required to assess the climate impacts on specific sectors, such as air quality and health. The need for state-of-the-art models is well illustrated by recent evidence of the important role of natural climate variability on regional atmospheric chemistry[76,77], further questioning the adequacy of damage functions based on global mean temperature. At the very least, the relevance of these damage functions could be tested against the more geospatially resolved and physically grounded CHES modeling framework.

The modeling framework presented in this study can offer a new and complementary way for multi-sectoral climate impact assessments under a wide range of up-to-date policy scenarios while ensuring the needed consistency among the various components of the human and Earth systems. We propose that the development of more integrated and self-consistent models of the coupled human and Earth systems, with a geospatially resolved physical representation of climate impacts be the next step beyond the traditional RCP and MIP approaches. Such an effort will promote an increasingly tighter collaboration among the IAM, ESM, and IAV communities. While there is still a need to bridge the gap between physical impacts and the resulting monetary values for economic damages, ongoing research shows important progress in this direction, such as efforts on health impacts[78,79] and agricultural impacts[80], and continued focus should be devoted on this aspect of climate impact research.

## Methods

**Coupled human-Earth system model.** In this study, we use the MIT Integrated Global System Modeling (IGSM) framework[14–17] that links a human system model, the economic projection and policy analysis (EPPA) model, to an ESM of intermediate complexity, the MIT Earth System Model (MESM). The schematic of the IGSM is provided in Supplementary Fig. 5.

**Human system model.** To evaluate long-term scenarios of energy and economic development we employ the EPPA model[81,82], which provides a multi-region, multi-sector dynamic representation of the global economy. The Global Trade Analysis Project (GTAP) dataset[83] provides the base information on the input-output structure for regional economies, including bilateral trade flows. The base year for the model is 2010, based on the calibration of the GTAP data for 2007, and from 2010 the model solves at 5-year intervals. We also further calibrate the data for 2010–2015 based on the data from the International Monetary Fund (IMF) World Economic Outlook[84] and the International Energy Agency (IEA) World Energy Outlook[85].

The model includes a representation of $CO_2$ and non-$CO_2$ ($CH_4$, $N_2O$, HFCs, PFCs, and $SF_6$) GHG emissions abatement, and calculates reductions from gas-specific control measures as well as those occurring as a byproduct of actions directed at reducing emissions of $CO_2$. The model also tracks major air pollutants: sulfates ($SO_x$), nitrogen oxides ($NO_x$), black carbon (BC), organic carbon (OC), carbon monoxide (CO), ammonia ($NH_3$), and non-methane volatile organic compounds (VOCs).

Future scenarios can be calibrated to specified energy or emissions profiles or driven by economic growth (resulting from savings and investments) and by exogenously specified productivity improvement in labor, energy, and land. Demand for goods produced from each sector increases as GDP and income grow; stocks of limited resources (e.g., coal, oil, and natural gas) deplete with use, driving production to higher cost grades; sectors that use renewable resources (e.g., land) compete for the available flow of services from them, generating rents. Combined

with policy and other constraints, these drivers change the relative economics of different technologies over time and across scenarios, as advanced technologies only enter the market when they become cost-competitive.

The production structure for electricity is the most detailed of all sectors, and captures technological changes that will be important to track under a GHG emissions mitigation policy. The deployment of advanced technologies is endogenous to the model. Advanced technologies, such as cellulosic biofuel or wind and solar technologies, enter the market when they become cost-competitive with existing technologies. Technologies are ranked according to their levelized cost of electricity, plus additional integration costs for wind and solar. When a carbon price exists, low carbon technologies are introduced. Initially, a fixed factor is required to represent costs of deployment (e.g., institutional costs, learning costs) for new technologies that—while competitive—require some time to penetrate into the market. The fixed-factor supply grows each period as a function of deployment until it becomes non-binding, allowing for large-scale deployment of the new technology. A complete description of the nesting structure of electricity generation and other production sectors in the EPPA model can be found in the model description[81].

**Earth system model.** The MESM[86] couples a zonally-averaged model of atmospheric dynamics, physics and chemistry, a land model with a representation of the terrestrial ecosystem biogeochemistry, and a choice of either a mixed layer anomaly diffusive ocean model or a 3-dimensional dynamical ocean component based on the MIT ocean general circulation model[87,88], including a detailed representation of physical, chemical, and biological processes[29–31], along with carbon cycle and thermodynamic sea-ice submodels.

The atmospheric model is a zonally-averaged statistical dynamical model that explicitly solves the primitive equations for the zonal mean state of the atmosphere and includes parameterizations of heat, moisture, and momentum transports by large-scale eddies based on baroclinic wave theory. The parameterizations of physical processes include clouds, convection, precipitation, radiation, boundary layer processes, and surface fluxes. The radiation code includes all significant GHGs ($H_2O$, $CO_2$, $CH_4$, $N_2O$, CFCs, and $O_3$) and eleven types of aerosols. The land model simulates terrestrial water, energy, carbon, and nitrogen budgets including carbon dioxide ($CO_2$) and trace gas emissions of methane ($CH_4$) and nitrous oxide ($N_2O$). The MESM also includes an urban air chemistry model and a detailed global scale zonal-mean atmospheric chemistry model that consider the chemical fate of 33 species, 41 gaseous-phase, and 12 aqueous-phase chemical reactions.

The global climate response of the MESM can be varied by modifying its climate sensitivity, strength of aerosol forcing and rate of ocean heat and carbon uptake, thus allowing for uncertainty analysis in global climate change. For regional studies, the MESM can be coupled to the NCAR 3-dimensional Community Atmosphere Model (CAM)[28] or to a climate emulator that relies on a pattern-scaling method that extends the MESM zonal mean variables based on climate change patterns from various climate models[26].

**Geospatially resolved physical representation of impacts.** To simulate the physical impacts of global change, the MIT IGSM is linked to a series of impact models. In this study, we focus on the representation of climate impacts on ocean acidification, air quality, water resources and agriculture.

Changes in pH are simulated using the MESM 3-dimensional dynamical ocean, with a detailed representation of physical, chemical, and biological processes. The MESM can simulate changes in ocean carbon uptake and acidification under various scenarios of global change, consistent with the associated changes in the physical ocean (e.g., warming and changes in the meridional overturning circulation). In addition, since the ocean model is fully coupled within the MESM, changes in ocean circulation and carbon impact the global climate system.

To estimate regional atmospheric pollutant concentrations, the IGSM is linked to GEOS-Chem version 9.02, a three-dimensional chemical transport model[89] with a 2° × 2.5° horizontal grid cell resolution. Non-agricultural anthropogenic emissions are projected in ten-year intervals out to 2100 using the projections from EPPA. In previous studies, we have coupled a three-dimensional chemical transport model to the MESM to examine the impact of climate change on air quality[77,79]. However, here, meteorological fields from 2010 were used, thus isolating the air quality impact of anthropogenic emissions changes, while biomass burning and biogenic emissions are left constant at 2010 levels.

We assess trends in water stress using the Water Resource System (WRS)[34,35], a river basin scale model of water resources management, which is forced by global simulations of climate change as well as socioeconomic drivers simulated by the IGSM. The WRS framework includes: (1) water supply: the collection, storage, and diversion of natural surface water and groundwater; (2) water requirements: the withdrawal, consumption, and flow management of water for economic and environmental purposes; and (3) the supply/requirement balance at river basin scale and measures of water scarcity. We assess changes in water stress for the globe at 282 Assessment Sub Regions (ASRs), which are geographic regions delineated by large river basin and country boundaries.

Finally, we estimate changes in agricultural productivity using the Terrestrial Ecosystem Model (TEM) component of the MESM, which is a process-based model that describes the carbon, nitrogen, and water dynamics of plants and soils for terrestrial ecosystems over the globe[38,90,91]. TEM uses spatially referenced

information on climate, elevation, soils, and vegetation as well as soil-specific and vegetation-specific parameters to estimate important carbon, nitrogen, and water fluxes and pool sizes of terrestrial ecosystems and land productivity for a large number of vegetation types, including crops. TEM has a 1-month time step and a 0.5° × 0.5° horizontal grid cell resolution. TEM is coupled to the EPPA model to provide an integrated modeling framework to project land-use change and its associated changes in land productivity and net land carbon fluxes[38,39,91], driven by changes in atmospheric carbon dioxide ($CO_2$) and ozone ($O_3$) concentrations and climate variables (i.e., temperature, precipitation, radiation) from the MESM model. In most studies, there is no feedback of land-use change GHG emissions and changes in albedo onto the climate system, however, the two-way coupling has been implemented for targeted studies[41].

**Code availability**. Various codes that support the findings of this study are publicly available. A public version of the EPPA 6 model can be downloaded upon request by emailing globalchangewebmaster@mit.edu. The MESM source code will be publicly available via repository once the user license is completed (email mesm-request@mit.edu for further information). The MITgcm source code is publicly available via repository at http://mitgcm.org. The versions of WRS and TEM models used in this study are maintained by the MIT Joint Program on the Science and Policy of Global Change and service requests should be directed to the corresponding authors. The GOES-Chem model is managed by the GEOS-Chem Support Team, based at Harvard University and Dalhousie University with support from the US NASA Earth Science Division and the Canadian National and Engineering Research Council and a public release of the model can be obtained at http://geos-chem.org/.

**Data availability**. The underlying data supporting the findings of the study are available at the DSpace@MIT (http://hdl.handle.net/1721.1/113296).

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

## Acknowledgements

The MIT Joint Program on the Science and Policy of Global Change is supported by an international consortium of government, industry and foundation sponsors. For a complete list, see https://globalchange.mit.edu/sponsors.

## Author contributions

E.M., S.P., A.S., C.A.S., R.P. and M.H. designed the research; E.M., S.P., A.S., Y.-H.H.C., X.G., Q.E., E.C., C.A.S., S.D., C.F., J.S., J.M. and D.K. performed the simulations and analyzed the data; and mainly E.M., S.P. and H.J. wrote the paper.

## Additional information

**Competing interests:** The authors declare no competing financial interests.

