## [Peer Review File · Nature Communications]

Editorial Note: This manuscript has been previously reviewed at another journal that is not operating a transparent peer review scheme. This document only contains reviewer comments and rebuttal letters for versions considered at Nature Communications. Mentions of prior referee reports have been redacted.

Reviewers' comments:

Reviewer #1 (Remarks to the Author):

The paper provides a valuable contribution to the science community in that it highlights an important development in the literature, namely the development of an integrated assessment modeling literature that includes detailed representations of human system derived climate forcers, atmosphere/climate, and geospatially resolved physical climate consequences.

To many people integrated assessment models (IAMs) are limited to a small group of highly aggregated models which have as their domain the social cost of carbon (SCC) and similar problems. Such models simplify the human and physical greatly. Another set of IAMs exist that have developed detailed representations of human and physical earth systems. The MIT EPPA model is a member of this latter group.

An important development in this latter IAM community is the development of capabilities to simultaneously explore emissions, atmosphere/climate, and climate consequences in a scientifically rigorous manner. More highly resolved IAMs have begun to develop a deeper literature focused on the development of internally consistent representations of impacts, adaptation and vulnerability (IAV) at higher spatial, temporal and sectoral resolution. This focus on providing higher resolution information relevant to IAV in addition to anthropogenic emissions and climate represents a paradigm shift in the higher-resolution IAM community. This paper does a nice job demonstrating the state of the art and clearly articulating this paradigm shift. It demonstrates the capability with four scenarios, one based on the nominal goal of limiting climate change to 2 degrees C relative to preindustrial, one based on the Paris Agreement's Nationally Determined Contributions (NDCs) without additional mitigation and two other policy scenarios, "Mountains" and "Oceans".

The four scenarios that the paper explores are not particularly interesting per se. There are numerous scenarios in the IPCC data base that preceded this work that provide anthropogenic emissions pathways that are similar. What is different about this paper is the fact that they are coupled with a suite of impacts that follow from the associated atmosphere/climate change. The virtue of this paper is that it reports on emissions, atmosphere/climate, and climate impacts and calls our attention to the new focus of the research community.

The paper does as less impressive job reflecting a knowledge of the broader research community's literature. The set of references is remarkably devoid of any inkling of research that was not conducted locally. Yet all of the major IAM groups around the world have made substantial contributions to the scientific literature exploring consistent scenarios of emissions, atmosphere/climate, and climate consequences in a scientifically rigorous manner, there is no recognition of this broader literature. This is a serious deficiency and needs to be rectified before publication.

Reviewer #2 (Remarks to the Author):

The paper "A paradigm shift toward a consistent modeling framework to assess climate impacts" aims at an important area in the integrated assessment of climate change, that is, the consistent inclusion of climate impacts. The modeling framework described offers an impressive depth of physical process detail.

My single most important critique is on the unclear positioning of this paper in between a methodological contribution and an application to projecting climate change impacts. The authors emphasize the methodological contribution in the motivation ("paradigm shift"), but most of the abstract and large parts of the main text are spend on exemplary results, which are, I think, not always new. Given length constraints it is impossible to deliver on both claims -- methodological innovation and application -- and the paper would greatly profit from concentrating more on one of those. I would suggest the methodological contribution to be the more interesting and appropriate for this journal.

Detailed review:

The most important claim of the paper is the development of a self-consistent human and earth system modeling framework of greater physical detail than existing model frameworks. Other claims relate to specific results on climate impacts.

Prior art includes IAM 1.0 and MIPs, as described by the authors, but beyond that also much more complex Integrated Assessment models that should be mentioned here: For example, integrated modeling frameworks of the energy system and local air pollution have been used by Rogelj et al. (doi:10.1038/nclimate2178) to assess air pollution. Klein et al. (doi:10.1007/s10584-013-0940-z) use an integrated energy-economy-land-use model to assess bioenergy and climate policy. While the authors write that the ESM, IAM, and IAV communities have remained separate, the two above mentioned examples are outcomes from very active collaborations across these communities. The IGSM model may have much more physical detail than these models, but the potential advantage over existing complex IAMs should be made more obvious. Also, novelty and advantages over previously published and referenced model versions of IGSM are not evident to me.

This work definitively contributes to the thinking about integrated assessment frameworks, but the practical use for scientists in the field is not made very clear: There is no detailed explanation of how the integration of all the different sub-models actually works and how 'self-consistency' is achieved. Furthermore, it would be of greatest interest for people in the field to know whether this model is openly available and the results thus reproducible by others, or published open source. Does this model require a supercomputer, or run on just a laptop?

The central claim of self-consistency of the model framework's sub-models is unfortunately hardly explained, and not demonstrated. I would guess self-consistency means the results of the model framework are equivalent to the results of a hypothetical model where all modeling detail is contained in a single model. The authors probably achieve this by soft-coupling the sub-models to each other, but this is not mentioned or explained at all. Demonstrating that physical simulations obtained with this model are close to what more detailed models, such as those in MIPs, would show, is left open. This paper would greatly benefit from a deeper discussion of the value added of self-consistency over existing integrated models and of the exact meaning and implementation of a self-consistency model coupling.

Specific results on climate impacts derived using IGSM are also presented in this paper, but the claims to novelty are mixed and not very clear to me: Climate impacts on crop yields and water stress are described by the authors as consistent with existing results from MIPs. By contrast, results on ocean acidification and local air pollution are presented without reference to prior art, and it is not evident to me whether the authors claim novelty of these results. Given the breadth of literature on these topics I would like to be convinced of the novelty by the authors.

In summary, I read this paper as "we have achieved a consistent integration of physical and process models to use as a basis to assess climate impacts on", which is in itself highly valuable. However, without a better description of how this is achieved (or publishing of the model itself), the value of publishing this work for others in the field is limited, and may rather warrant the format of a commentary.

Specific comments:

- "Under the 2C scenario, concentrations in PM 2.5 decrease compared to present day, largely due to co-benefits from reductions in co-emitted pollutants from imposing a carbon tax": I do not understand this sentence: Isn't the reduced local air pollution itself a co-benefit of the climate policy (as a physical by-effect of climate policy)? The sentence indicates that this reduction in pollution is caused by co-benefits, which implies some kind of valuation of the human impacts of pollution -- which are not part of this model to my reading of this paper.

- ".. and avoid reliance on massive unproven negative emissions technologies, like the RCP2.6." But this study does include a 2C scenario. Is it less stringent than the RCP2.6 (which is likely to stay below

2 degree), or does it explicitly exclude negative emissions technologies?

- the acronym IGSM is never defined in the main text

- the link to the model description Paltsev et al. (2005) is broken. In fact, all the http://globalchange.mit.edu/files/document/* references I tried were broken.

Reviewer #3 (Remarks to the Author):

In general, I like the type of analysis proposed in this paper and agree with the authors that this is the direction the research community needs to go. However, I think the authors need to strengthen this argument in the article. In particular, I suggest interweaving the reasons IAM 1.0 doesn't work within the discussion of the sectoral results. In each instance, noting why the old way would provide incorrect results. Furthermore, I don't think the authors make the convincing case for the particular configuration they seem to argue for (EMIC). Why wouldn't a different representation of the Earth system work? I also think the authors need to more clearly differentiate from previous literature, in particular the references I include in this report.

Specific Comments:

1) The first sentence is a little misleading. Most of the IAMs that are in the IPCC either exclude impacts or do a detailed one-way study on a particular sector (see reference list below). The models you seem to be referring to are DICE, FUND, PAGE. The characterization of them is fair, but it is worth noting that there are other models that can do detailed impacts. Some of these models are also pursuing IAM 2.0 frameworks.

2) Lines 105-109: if your results are consistent with AgMIP, why do you need this approach?

3) What do you lose by using an EMIC over an ESM? What do you gain by using the EMIC over the box model? I think these trade-offs will be important to understand.

4) Lines 152-154: it is important to note that your proposal could help quantify parametric or scenario uncertainty, but to address structural uncertainty you would need either multiple models or multiple model components.

5) It seems to me that the structure you propose may address some concerns with previous methods, but still wouldn't be a replacement for the damage functions produced by DICE, FUND, PAGE since you are only looking at a small number of sectors. What do you do about impacts excluded from your model?

6) There are other papers that discuss these sorts of frameworks, either itemizing the pros & cons (van Vuuren et al., 2012) or describing a particular system (Collins et al., 2015). These should be included in this discussion.

7) I don't think I fully understand your choice in scenarios. It seems that the point you want to make could be made with any scenario, and would probably be the most clearly articulated with a high & low scenario. What do Mountains & Oceans add? And, is it important that you exclude an RCP8.5 like scenario?

8) Figure 4 isn't very helpful. It would probably be more informative to structure it like Figure 3 or 2, where we can see results under multiple scenarios.

9) How are the modules linked? Do impacts influence emissions & climate?

References for comment #1:

Isaac, M. and D. P. van Vuuren (2009). "Modeling global residential sector energy demand for heating and air conditioning in the context of climate change." *Energy Policy* 37: 507-521.

Labriet, M., S. Joshi, F. Babonneau, N. Edwards, P. Holden, A. Kanudia, R. Loulou and M. Vielle (2013). "Worldwide impacts of climate change on energy for heating and cooling." *Mitigation and Adaptation Strategies for Global Change*.

Zhou, Y., J. Eom and L. Clarke (2013). "The effect of global climate change, population distribution, and climate mitigation on building energy use in the U.S. and China." *Climatic Change* 119(3): 979-992.

Kyle, P., C. Mueller, K. Calvin and A. M. Thomson (2014). "Meeting the Radiative Forcing Targets of the Representative Concentration Pathways in a World with Agricultural Climate Impacts." *Earths Future* 2(2): 83-98.

Nelson, G., D. van der Mensbrugghe, E. Blanc, K. Calvin, T. Hasegawa, P. Havlik, P. Kyle, H. Lotze-Campen, M. von Lampe, D. Mason d'Croze, H. van Meijl, C. Mueller, J. Reilly, R. Robertson, R. Sands, C. Schmitz, A. Tabeau, K. Takahashi and H. Valin (2014). "Agriculture and climate change in global scenarios: why don't the models agree." *Agricultural Economics* 45(1).

Reilly, J., S. Paltsev, B. Felzer, X. Wang, D. Kicklighter, J. Melillo, R. Prinn, M. Sarofim, A. Sokolov and C. Wang (2007). "Global economic effects of changes in crops, pasture, and forests due to changing climate, carbon dioxide, and ozone." *Energy Policy* 35: 5370-5383.

Hanasaki, N., S. Fujimori, T. Yamamoto, S. Yoshikawa, Y. Masaki, Y. Hijioka, M. Kainuma, Y. Kanamori, T. Masui, K. Takahashi and S. Kanae (2013). "A global water scarcity assessment under Shared Socio-economic Pathways -- Part 2: Water availability and scarcity." *Hydrological Earth System Science* 17: 2393-2413.

Kim, S. H., M. Hejazi, L. Liu, K. Calvin, L. Clarke, J. Edmonds, P. Kyle, P. Patel, M. Wise and E. Davies (2016). "Balancing global water availability and use at basin scale in an integrated assessment model." *Climatic Change* 136(2): 217-231.

Hejazi, M., J. Edmonds, L. Clarke, P. Kyle, E. Davies, V. Chaturvedi, M. Wise, P. Patel, J. Eom and K. Calvin (2014). "Integrated assessment of water scarcity over the 21st century under multiple climate change mitigation policies." *Hydrology and Earth System Sciences* 18: 2859-2883.

Schlösser, A., K. Strzepek, X. Gao, C. Fant, E. Blanc, S. Paltsev, H. Jacoby, J. Reilly and Gueneau (2014). "The future of global water stress: An integrated assessment." *Earths Future* 2(8): 341-361.

Mima, S. and P. Criqui (2009). Assessment of the impacts under future climate change on the energy systems with the POLES model. 2009 International Energy Workshop, Venice, Italy.

References for comment #6:

van Vuuren, D. P., L. Bayer, C. Chuwah, L. Ganzeveld, W. Hazeleger, B. van den Hurk, T. van Noije, B. O'Neill and B. Strengers (2012). "A comprehensive view on climate change coupling of earth system and integrated assessment models." *Environmental Research Letters* 7(024012).

Collins, W. D., A. P. Craig, J. E. Truesdale, A. V. Di Vittorio, A. D. Jones, B. Bond-Lamberty, K. V. Calvin, J. A. Edmonds, S. H. Kim, A. M. Thomson, P. Patel, Y. Zhou, J. Mao, X. Shi, P. E. Thornton, L. P. Chini and G. C. Hurtt (2015). "The integrated Earth system model version 1: formulation and functionality." *Geosci. Model Dev.* 8(7): 2203-2219.

We thank the reviewers for their constructive comments and suggestions, which we believe have helped us improve the manuscript. We provide point-by-point responses to their comments below. Overall, we followed the suggestion from the editor and restructured the manuscript with a focus on the methodological advance, discussing how various physical impact models are integrated, the coupling strategies, and giving only a brief showcase application to provide concrete examples of the potential capability of the proposed coupled human and Earth system modeling framework. We tried best to address all the reviewers' concerns while keeping the word count reasonable. We hope it addresses the main concerns of the reviewers.

Reviewers' comments:

Reviewer #1 (Remarks to the Author):

The paper provides a valuable contribution to the science community in that it highlights an important development in the literature, namely the development of an integrated assessment modeling literature that includes detailed representations of human system derived climate forcers, atmosphere/climate, and geospatially resolved physical climate consequences.

To many people integrated assessment models (IAMs) are limited to a small group of highly aggregated models which have as their domain the social cost of carbon (SCC) and similar problems. Such models simplify the human and physical greatly. Another set of IAMs exist that have developed detailed representations of human and physical earth systems. The MIT EPPA model is a member of this latter group.

An important development in this latter IAM community is the development of capabilities to simultaneously explore emissions, atmosphere/climate, and climate consequences in a scientifically rigorous manner. More highly resolved IAMs have begun to develop a deeper literature focused on the development of internally consistent representations of impacts, adaptation and vulnerability (IAV) at higher spatial, temporal and sectoral resolution. This focus on providing higher resolution information relevant to IAV in addition to anthropogenic emissions and climate represents a paradigm shift in the higher-resolution IAM community. This paper does a nice job demonstrating the state of the art and clearly articulating this paradigm shift. It demonstrates the capability with four scenarios, one based on the nominal goal of limiting climate change to 2 degrees C relative to preindustrial, one based on the Paris Agreement's Nationally Determined Contributions (NDCs) without additional mitigation and two other policy scenarios, "Mountains" and "Oceans".

We appreciate the reviewer's understanding of the main objectives of our manuscript.

The four scenarios that the paper explores are not particularly interesting per se. There are numerous scenarios in the IPCC data base that preceded this work that provide anthropogenic emissions pathways that are similar. What is different about this paper is the fact that they are coupled with a suite of impacts that follow from the associated atmosphere/climate change. The virtue of this paper is that it reports on emissions,

atmosphere/climate, and climate impacts and calls our attention to the new focus of the research community.

We agree with the reviewer and make this point more clearly in the revised manuscript by adding the following sentence in the revised manuscript: “few studies bring together estimates of climate impacts across sectors of the economy under a consistent modeling framework and consistent socio-economic and climate scenarios”

The paper does as less impressive job reflecting a knowledge of the broader research community’s literature. The set of references is remarkably devoid of any inkling of research that was not conducted locally. Yet all of the major IAM groups around the world have made substantial contributions to the scientific literature exploring consistent scenarios of emissions, atmosphere/climate, and climate consequences in a scientifically rigorous manner, there is no recognition of this broader literature. This is a serious deficiency and needs to be rectified before publication.

In the revised manuscript, we make two major additions to respond to that comment:

- we include a more comprehensive review of substantial contributions to the scientific literature by other IAM groups;
- we make clear that the MIT IGSM framework is only used as an example of the potential capabilities of improved coupled human and Earth system models, and that other IAMs have made similar improvements in the integration of geospatially resolved physical climate and thus could easily be used instead of the IGSM.

We hope these revisions address the reviewer’s concerns and substantially improve the manuscript.

Reviewer #2 (Remarks to the Author):

The paper "A paradigm shift toward a consistent modeling framework to assess climate impacts" aims at an important area in the integrated assessment of climate change, that is, the consistent inclusion of climate impacts. The modeling framework described offers an impressive depth of physical process detail.

My single most important critique is on the unclear positioning of this paper in between a methodological contribution and an application to projecting climate change impacts. The authors emphasize the methodological contribution in the motivation ("paradigm shift"), but most of the abstract and large parts of the main text are spend on exemplary results, which are, I think, not always new. Given length constraints it is impossible to deliver on both claims -- methodological innovation and application -- and the paper would greatly profit from concentrating more on one of those. I would suggest the methodological contribution to be the more interesting and appropriate for this journal.

We agree with the reviewer that we attempted, apparently not successfully, to include both a methodological contribution and an application to projecting climate change impacts. In our revisions, we restructured the manuscript with a stronger focus on the methodological advance, discussing how various physical impact models are integrated, the coupling strategies, and the improved responsiveness, flexibility and consistency of the new modeling framework. Nonetheless, we still offer a brief showcase application to provide concrete examples of the potential capability of the proposed coupled human and Earth system modeling framework manuscript, so that the manuscript is not just conceptual. We hope this will satisfy the reviewer's concerns.

Detailed review:

The most important claim of the paper is the development of a self-consistent human and earth system modeling framework of greater physical detail than existing model frameworks. Other claims relate to specific results on climate impacts.

Prior art includes IAM 1.0 and MIPs, as described by the authors, but beyond that also much more complex Integrated Assessment models that should be mentioned here: For example, integrated modeling frameworks of the energy system and local air pollution have been used by Rogelj et al. (doi:10.1038/nclimate2178) to assess air pollution. Klein et al. (doi:10.1007/s10584-013-0940-z) use an integrated energy-economy-land-use model to assess bioenergy and climate policy.

While the authors write that the ESM, IAM, and IAV communities have remained separate, the two above mentioned examples are outcomes from very active collaborations across these communities. The IGSM model may have much more physical detail than these models, but the potential advantage over existing complex IAMs should be made more obvious. Also, novelty and advantages over previously published and referenced model versions of IGSM are not evident to me.

In the revised manuscript, we acknowledge the improvements in many integrated modeling frameworks, by including a more comprehensive review of substantial contributions to the scientific literature by other IAM groups, including the two mentioned by the reviewer. We also recognize that, while the ESM, IAM and IAV communities generally remain separate, they do actively collaborate.

This work definitively contributes to the thinking about integrated assessment frameworks, but the practical use for scientists in the field is not made very clear: There is no detailed explanation of how the integration of all the different sub-models actually works and how 'self-consistency' is achieved. Furthermore, it would be of greatest interest for people in the field to know whether this model is openly available and the results thus reproducible by others, or published open source. Does this model require a supercomputer, or run on just a laptop?

The various impact models that are integrated into the coupled human and Earth system modeling framework have different levels of computational demand. We discuss the use of emulators and the development of computationally efficient

framework for risk analysis. We also discuss the need for computational demanding process-based to assess the impacts of specific sectors, like air quality and health.

We are currently working on a user license with the MIT Technology Licensing Office and thus are hoping to release a version of the IGSM to the public in the months to come.

The central claim of self-consistency of the model framework's sub-models is unfortunately hardly explained, and not demonstrated. I would guess self-consistency means the results of the model framework are equivalent to the results of a hypothetical model where all modeling detail is contained in a single model. The authors probably achieve this by soft-coupling the sub-models to each other, but this is not mentioned or explained at all. Demonstrating that physical simulations obtained with this model are close to what more detailed models, such as those in MIPs, would show, is left open. This paper would greatly benefit from a deeper discussion of the valued added of self-consistency over existing integrated models and of the exact meaning and implementation of a self-consistency model coupling.

First, we provide more details about the inconsistency of the existing design of impact assessments. Second, we add a discussion of coupling strategies and further details on the level of integration of the various impact models considered in our analysis.

Specific results on climate impacts derived using IGSM are also presented in this paper, but the claims to novelty are mixed and not very clear to me: Climate impacts on crop yields and water stress are described by the authors as consistent with existing results from MIPs. By contrast, results on ocean acidification and local air pollution are presented without reference to prior art, and it is not evident to me whether the authors claim novelty of these results. Given the breadth of literature on these topics I would like to be convinced of the novelty by the authors.

In our view, all results we present are novel and some of them are consistent with existing estimates from very detailed exercises, but we've made major revisions so that the focus of this paper is now on the methodology, with sample results illustrating the approach (which happens to be novel). In addition, we highlight that, to our knowledge, very few studies bring together estimates of climate impacts across ecosystems and sectors of the economy under a consistent modeling framework and consistent socio-economic and climate scenarios.

In summary, I read this paper as "we have achieved a consistent integration of physical and process models to use as a basis to assess climate impacts on", which is in itself highly valuable. However, without a better description of how this is achieved (or publishing of the model itself), the value of publishing this work for others in the field is limited, and may rather warrant the format of a commentary.

To address this comment, we provide more details on how the geospatially resolved physical impact models are integrated within the IGSM and discuss coupling strategies.

Specific comments:

- "Under the 2C scenario, concentrations in PM 2.5 decrease compared to present day, largely due to co-benefits from reductions in co-emitted pollutants from imposing a carbon tax": I do not understand this sentence: Isn't the reduced local air pollution itself a co-benefit of the climate policy (as a physical by-effect of climate policy)? The sentence indicates that this reduction in pollution is caused by co-benefits, which implies some kind of valuation of the human impacts of pollution -- which are not part of this model to my reading of this paper.

We changed the sentence to: "would experience significant decreases in PM_{2.5} under the 2C scenario as a co-benefit from imposing a carbon tax". Hopefully that clarifies the statement, which the reviewer understood correctly (the reduced local air pollution itself is a co-benefit of the climate policy).

- "... and avoid reliance on massive unproven negative emissions technologies, like the RCP2.6." But this study does include a 2C scenario. Is it less stringent than the RCP2.6 (which is likely to stay below 2 degree), or does it explicitly exclude negative emissions technologies?

It is less stringent than the RCP2.6 and does not require negative emissions technologies.

- the acronym IGSM is never defined in the main text

We corrected that oversight.

- the link to the model description Paltsev et al. (2005) is broken. In fact, all the <http://globalchange.mit.edu/files/document/>* references I tried were broken.

During the review process for this manuscript, we had a update of our program's website and the URL to our reports were changed. We updated the links in the references.

We hope the revised manuscript addresses the reviewer's concerns, which is difficult given the need to keep the manuscript to a reasonable length.

Reviewer #3 (Remarks to the Author):

In general, I like the type of analysis proposed in this paper and agree with the authors that this is the direction the research community needs to go. However, I think the authors need to strengthen this argument in the article. In particular, I suggest

interweaving the reasons IAM 1.0 doesn't work within the discussion of the sectoral results. In each instance, noting why the old way would provide incorrect results. Furthermore, I don't think the authors make the convincing case for the particular configuration they seem to argue for (EMIC). Why wouldn't a different representation of the Earth system work? I also think the authors need to more clearly differentiate from previous literature, in particular the references I include in this report.

Given the concerns from the other reviewers and the suggestion by the editor to focus more on the methodology and less on the actual climate impact assessment, it is difficult to "interweaving the reasons IAM 1.0 doesn't work within the discussion of the sectoral results" and "In each instance, noting why the old way would provide incorrect results".

As for the particular configuration of the Earth system we rely on, we make clear that it is not the only or even preferred strategy, referencing Collins et al. (2015), and discussing using actual climate models versus climate emulator techniques.

Specific Comments:

1) The first sentence is a little misleading. Most of the IAMs that are in the IPCC either exclude impacts or do a detailed one-way study on a particular sector (see reference list below). The models you seem to be referring to are DICE, FUND, PAGE. The characterization of them is fair, but it is worth noting that there are other models that can do detailed impacts. Some of these models are also pursuing IAM 2.0 frameworks.

In the revised manuscript, we acknowledge the development to the next generation of integrated coupled human and Earth systems by various IAM groups and we provide a more comprehensive review of substantial contributions to the scientific literature by other IAM groups, including many of the suggested references (not all solely because of the limit on the number of references).

2) Lines 105-109: if your results are consistent with AgMIP, why do you need this approach?

Our approach is needed because it is consistent not only with the impacts studied by AgMIP, and other ISIMIP impact assessment, but also with other aspects of human-Earth interactions (e.g, impacts of air pollution). And since it doesn't require cumbersome internationally coordinated research exercises involving different research communities and multiple modeling groups, it provides a flexible modeling framework that is responsive to evolving global climate policy. We hope this is made clear in the revised manuscript.

3) What do you lose by using an EMIC over an ESM? What do you gain by using the EMIC over the box model? I think these trade-offs will be important to understand.

Given the word constraints, we limited our discussion to the use of statistical emulation techniques of 3-dimensional climate models versus (pattern scaling).

4) Lines 152-154: it is important to note that your proposal could help quantify parametric or scenario uncertainty, but to address structural uncertainty you would need either multiple models or multiple model components.

We added the following sentence: “Such approach could not only help quantify parametric or scenario uncertainty, but also address structural uncertainty by using emulators to represent multiple models or multiple model components”.

5) It seems to me that the structure you propose may address some concerns with previous methods, but still wouldn't be a replacement for the damage functions produced by DICE, FUND, PAGE since you are only looking at a small number of sectors. What do you do about impacts excluded from your model?

We added the following sentence in the revised manuscript: “the relevance of these damage functions could be tested against the more geospatially resolved and physically grounded CHES modeling framework.”

6) There are other papers that discuss these sorts of frameworks, either itemizing the pros & cons (van Vuuren et al., 2012) or describing a particular system (Collins et al., 2015). These should be included in this discussion.

The reviewer makes a very good point. In the revised manuscript, we provide a brief description of coupling strategies and level of integration between the human system and the Earth system, referencing both van Vuuren et al. (2012) and Collins et al. (2015).

7) I don't think I fully understand your choice in scenarios. It seems that the point you want to make could be made with any scenario, and would probably be the most clearly articulated with a high & low scenario. What do Mountains & Oceans add? And, is it important that you exclude an RCP8.5 like scenario?

We use the scenarios to demonstrate the capability and responsiveness of the modeling framework to design scenarios using typical approaches to deal with the future in business, government and academia. But we agree with the reviewer and focus largely on the high and low scenario in our sample impact assessment, adding the analysis of the other scenarios in the supplemental materials.

8) Figure 4 isn't very helpful. It would probably be more informative to structure it like Figure 3 or 2, where we can see results under multiple scenarios.

We redid the analysis of the water resources and agriculture impacts to be more clear and straightforward, focusing mainly on the *Paris Forever* and *2C* scenarios.

9) How are the modules linked? Do impacts influence emissions & climate?

We provide a discussion of coupling strategies and then indicate more specifically the level of coupling of the various impact models.

We hope these major revisions address the reviewer's concerns, and at the very least the most important ones. Given the need to keep the manuscript to a reasonable length, we tried our best to strike the right balance in our revisions, hopefully with success.

References for comment #1:

Isaac, M. and D. P. van Vuuren (2009). "Modeling global residential sector energy demand for heating and air conditioning in the context of climate change." *Energy Policy* 37: 507-521.

Labriet, M., S. Joshi, F. Babonneau, N. Edwards, P. Holden, A. Kanudia, R. Loulou and M. Vielle (2013). "Worldwide impacts of climate change on energy for heating and cooling." *Mitigation and Adaptation Strategies for Global Change*.

Zhou, Y., J. Eom and L. Clarke (2013). "The effect of global climate change, population distribution, and climate mitigation on building energy use in the U.S. and China." *Climatic Change* 119(3): 979-992.

Kyle, P., C. Mueller, K. Calvin and A. M. Thomson (2014). "Meeting the Radiative Forcing Targets of the Representative Concentration Pathways in a World with Agricultural Climate Impacts." *Earths Future* 2(2): 83-98.

Nelson, G., D. van der Mensbrugge, E. Blanc, K. Calvin, T. Hasegawa, P. Havlik, P. Kyle, H. Lotze-Campen, M. von Lampe, D. Mason d'Croze, H. van Meijl, C. Mueller, J. Reilly, R. Robertson, R. Sands, C. Schmitz, A. Tabeau, K. Takahashi and H. Valin (2014). "Agriculture and climate change in global scenarios: why don't the models agree." *Agricultural Economics* 45(1).

Reilly, J., S. Paltsev, B. Felzer, X. Wang, D. Kicklighter, J. Melillo, R. Prinn, M. Sarofim, A. Sokolov and C. Wang (2007). "Global economic effects of changes in crops, pasture, and forests due to changing climate, carbon dioxide, and ozone." *Energy Policy* 35: 5370-5383.

Hanasaki, N., S. Fujimori, T. Yamamoto, S. Yoshikawa, Y. Masaki, Y. Hijioka, M. Kainuma, Y. Kanamori, T. Masui, K. Takahashi and S. Kanae (2013). "A global water scarcity assessment under Shared Socio-economic Pathways -- Part 2: Water availability and scarcity." *Hydrological Earth System Science* 17: 2393-2413.

Kim, S. H., M. Hejazi, L. Liu, K. Calvin, L. Clarke, J. Edmonds, P. Kyle, P. Patel, M. Wise and E. Davies (2016). "Balancing global water availability and use at basin scale

in an integrated assessment model." *Climatic Change* 136(2): 217-231.

Hejazi, M., J. Edmonds, L. Clarke, P. Kyle, E. Davies, V. Chaturvedi, M. Wise, P. Patel, J. Eom and K. Calvin (2014). "Integrated assessment of water scarcity over the 21st century under multiple climate change mitigation policies." *Hydrology and Earth System Sciences* 18: 2859-2883.

Schlosser, A., K. Strzepek, X. Gao, C. Fant, E. Blanc, S. Paltsev, H. Jacoby, J. Reilly and Gueneau (2014). "The future of global water stress: An integrated assessment." *Earths Future* 2(8): 341-361.

Mima, S. and P. Criqui (2009). Assessment of the impacts under future climate change on the energy systems with the POLES model. 2009 International Energy Workshop, Venice, Italy.

References for comment #6:

van Vuuren, D. P., L. Bayer, C. Chuwah, L. Ganzeveld, W. Hazeleger, B. van den Hurk, T. van Noije, B. O'Neill and B. Strengers (2012). "A comprehensive view on climate change coupling of earth system and integrated assessment models." *Environmental Research Letters* 7(024012).

Collins, W. D., A. P. Craig, J. E. Truesdale, A. V. Di Vittorio, A. D. Jones, B. Bond-Lamberty, K. V. Calvin, J. A. Edmonds, S. H. Kim, A. M. Thomson, P. Patel, Y. Zhou, J. Mao, X. Shi, P. E. Thornton, L. P. Chini and G. C. Hurtt (2015). "The integrated Earth system model version 1: formulation and functionality." *Geosci. Model Dev.* 8(7): 2203-2219.

Reviewers' comments:

Reviewer #1 (Remarks to the Author):

The authors have added acknowledgement of similar work going on throughout the community of complex human-Earth system modelers. While the review of that body of work is still quite thin, I think that paper's main point is of sufficient merit that it warrants publication without any additional changes.

Reviewer #2 (Remarks to the Author):

Thank you very much for the revisions, I think the paper has a much clearer focus now.

However, I still think that the methodological innovation is not communicated clearly enough. Take for example this sentence on the methodology, which is a central part to the claim of novelty:

"Different coupling strategies exist, such as off-line one-way information exchange, online one-way exchange of

information from the human system to the Earth system or from the Earth system to the human system, and full online coupling with more or less instantaneous two-way interactions."

I do not understand, neither is it explained anywhere, what online vs. offline coupling means or what a "more or less instantaneous" interaction between models is. To my understanding, one can couple two models "one-shot" (taking results from one model, running the other model with this as an input), or iteratively include the feedbacks between the models. The second version is what I would consider a consistent integration.

How does this relate to your language of one-way, two-way, full, online, offline, and consistent coupling?

Given the centrality of this argument, this has to be explained in more detail, at least in the supplements.

I have a couple of other points:

- You mention that the 2C scenario is implemented as a carbon tax, but what about the other scenarios? What are the policy instruments in the other scenarios, e.g. what is the policy instrument by which "a large role for renewables" is implemented?

- I think the scenario naming "Paris forever" is very misleading. The Paris Agreement's aim is to limit temperature rise to "well below 2C". Your "Paris forever" scenario comes out at 3.6C in 2100. If the temperatures in Fig. 2 are the best-guess outcomes (so for a median ECS or TCRE), I could see how a RCP2.6 is "well below 2C". Your 2C scenario is significantly above that. I understand that you interpret the current Nationally Determined Contributions as the outcome of the Paris Agreement, but if so, that should be distinguished from the aim of the Paris Agreement.

- "human activity model" vs. "human system model" vs. "economic model" -- are these all the same?

The manuscript would profit much from a thorough editing and more concise language.

I found it hard to follow some long and convoluted sentences, e.g.

"Under various levels of climate mitigation, reductions in surface ozone concentrations, especially by 2100, along with reduced warming, while accounting for land-use change adaptation can result in benefits to the agriculture sector."

More specifically, referring to line numbers in the main text:

- 63-66: ungrammatical sentence

- 78/79: ungrammatical sentence

- 128: needs a "the"
- 151: Vague language: "using a different climate model" cannot "have a major impact on the effect of climate mitigation". Climate models may show large uncertainty in simulated impacts -- if that is what is meant, I think it would be best to say so in clear terms.
- 166: varying
- 177: cannot make sense of the ending of this sentence
- 177-179: ungrammatical and ambiguous
- 190-192: ungrammatical
- 217-219: ungrammatical
- 223-227: very long and ungrammatical sentence
- 231: wide range
- 231: What are "responsive policy scenarios"?
- 183-185: This is a very bleak take on the state of economic assessment of climate impacts. There is much and very active progress on this front, and you mention some counterexamples further below in the text.

Reviewer #3 (Remarks to the Author):

The authors have done a nice job addressing my comments and the comments of the other reviewers. I have a few small, editorial comments remaining:

Line 33: Should be "it results" instead of "its results"

Lines 144-149: Sentence is long and confusing, consider revising

Line 223: "computationally demanding"?

We thank the reviewers for their constructive comments and suggestions and we provide point-by-point responses to their comments below. In addition to revising the manuscript based on the reviewers' comments, we also reformatted the manuscript to match the requirements of *Nature Communications*, such as adding sections (introduction, results and discussions) and incorporating the methods section (originally a supplementary document). We also completed a thorough editing of the manuscript to improve its conciseness and clarity.

Reviewers' comments:

Reviewer #1 (Remarks to the Author):

The authors have added acknowledgement of similar work going on throughout the community of complex human-Earth system modelers. While the review of that body of work is still quite thin, I think that paper's main point is of sufficient merit that it warrants publication without any additional changes.

We thank the reviewer for helping us improve our manuscript.

Reviewer #2 (Remarks to the Author):

Thank you very much for the revisions, I think the paper has a much clearer focus now.

However, I still think that the methodological innovation is not communicated clearly enough. Take for example this sentence on the methodology, which is a central part to the claim of novelty: "Different coupling strategies exist, such as off-line one-way information exchange, online one-way exchange of information from the human system to the Earth system or from the Earth system to the human system, and full online coupling with more or less instantaneous two-way interactions."

I do not understand, neither is it explained anywhere, what online vs. offline coupling means or what a "more or less instantaneous" interaction between models is. To my understanding, one can couple two models "one-shot" (taking results from one model, running the other model with this as an input), or iteratively include the feedbacks between the models. The second version is what I would consider a consistent integration.

How does this relate to your language of one-way, two-way, full, online, offline, and consistent coupling?

Given the centrality of this argument, this has to be explained in more detail, at least in the supplements.

We extended and improved the discussion on coupling strategies to the following two paragraphs: "In practice, because state-of-the-art Earth system models are computationally expensive, a CHES model can be built by coupling a human system model to a simplified model of the climate system and to specific impact

models for key ecosystems and sectors of the economy (Fig. 1). Different coupling strategies exist¹⁴, from off-line one-way information exchange between research communities to fully coupled modeling approaches that yield more or less instantaneous (depending on the timestep of the coupling) two-way interactions between the human and Earth system components. Other strategies include improving the representation of the Earth system in IAMs or improving the representation of societal elements within ESMs. Beyond the challenge of coupling the human and Earth systems, an important characteristic of CHES models should be detailed representation of the biophysical impacts of climate change, spanning key economic sectors and ecosystem services. (Additional details on the various coupling strategies and their advantages are provided in the Supplementary Table 1.)

While the coupling strategy remains an important issue, and full coupling between the human and Earth systems is an aspirational goal, the effort will not be very insightful if it involves dubious damage functions, like it does in social cost of carbon models^{15,16}. Also, full coupling raises many additional challenges, e.g., difficulties of coupling different software systems, complexities of representing the cascading of uncertainty among components of the system, and differences in temporal and spatial scale of the various components. As a result, full coupling is generally limited to a specific pathway, like the land system¹⁷. In addition, full integration is not warranted unless there is evidence that it would substantially change the estimates of climate impacts. In the process of developing a CHES model, therefore, a one-way coupling where physical impacts of climate change are explicitly modeled, but do feed back onto emissions of greenhouse gases and the climate system (e.g., land-use change), is a useful first step. A salient response from the one-way testing will then warrant exploration of two-way coupling which, if found to produce significant new insights, can be incorporated in subsequent versions of the model. A similar approach is suggested to interactions among impact models, for example between air quality and agriculture

We also added a table in the supplementary information summarizing the various coupling strategies and their advantages.

I have a couple of other points:

- You mention that the 2C scenario is implemented as a carbon tax, but what about the other scenarios? What are the policy instruments in the other scenarios, e.g. what is the policy instrument by which "a large role for renewables" is implemented?

We provide more details in the Supplementary Table 1. The following text is added "Renewables are supported by renewables portfolio standards" and "Energy projections in *Oceans* and *Mountains* scenarios represent the view of the industry (developed by the Shell scenarios team) and were implemented in the EPPA model (the human system model of the MIT IGSM) by calibrating the total primary energy use on a regional basis".

We would be happy to move the Supplementary Table 1 into the main body if you feel it can clarify the description of the scenarios.

- I think the scenario naming "Paris forever" is very misleading. The Paris Agreement's aim is to limit temperature rise to "well below 2C". Your "Paris forever" scenario comes out at 3.6C in 2100. If the temperatures in Fig. 2 are the best-guess outcomes (so for a median ECS or TCRE), I could see how a RCP2.6 is "well below 2C". Your 2C scenario is significantly above that. I understand that you interpret the current Nationally Determined Contributions as the outcome of the Paris Agreement, but if so, that should be distinguished from the aim of the Paris Agreement.

We agree about the ambiguity of the term "Paris Forever, therefore we added the following detail "An outlook based on existing policy, here an assessment of the results from the UN COP-21 meeting⁴² (named Paris Forever), assuming no additional climate policy after 2030, resulting in 3.5°C warming in 2100, emphasizing that the current pledges are not sufficient to meet the goal to stay "well below 2°C"."

We also modified the Supplementary Table 1 to state the following to distinguish between the current NDCs and the aim of the Paris agreement:

"The *Paris Forever* scenario shows how far the current emission pledges take us since there is no agreement on the emission mitigation trajectories after 2030. We do not impose any additional climate policy after 2030 to illustrate that the current pledges are not enough to meet to goal to stay "well below 2°C"."

Fawcett et al (2015) investigate the outcome of the Paris agreement on increases in global mean temperature by 2100 using scenarios with different assumptions about emissions after 2030. Their analysis includes two scenarios named "Paris–Continued ambition" and "Paris–Increased ambition", having respectively 8% and 30% probability of staying below 2C and thus neither guarantee temperature increases well below 2C. Here we also make assumptions about what climate mitigation efforts past 2030, in this case assuming no additional climate policy past 2030.

We believe that this explanation will make sure that the reader will understand the difference between the current pledges and the ultimate goal of the Paris agreement.

See: Fawcett A and co-authors (2015) Can Paris pledges avert severe climate change? *Science*, **350**, 1168-1169, doi: 10.1126/science.aad5761

- "human activity model" vs. "human system model" vs. "economic model" -- are these all the same?

We realize the lack of consistency and have used “human system model” throughout the manuscript.

The manuscript would profit much from a thorough editing and more concise language. I found it hard to follow some long and convoluted sentences, e.g. "Under various levels of climate mitigation, reductions in surface ozone concentrations, especially by 2100, along with reduced warming, while accounting for land-use change adaptation can result in benefits to the agriculture sector."

We thank the reviewer for bringing to our attention the need for a thorough editing and more concise language. We have made a substantial effort to edit the manuscript and we hope the reviewer will find it much improved.

For example, the sentence mentioned, along with the previous one, are simplified and improved: “Finally, large increases in temperature, exceeding the damaging temperature thresholds for crop productivity, and major ozone damage are projected under the Paris Forever scenario. Even under cropland relocation, extension and intensification, the overall global crop yield (over crop land areas) decreases by 2100. Climate mitigation results in substantial reductions in warming and surface ozone concentrations, so land-use change adaptation can lead to benefits to the agriculture sector.”

More specifically, referring to line numbers in the main text:

- 63-66: ungrammatical sentence

We split the sentence in two: “The human system component of a CHES model should represent the world’s economy, disaggregated into multiple regions and with sectoral detail (i.e., agriculture, services, industrial and household transportation, energy-intensive industry). It also should include trade, investments, savings, and consumption decisions, as well as abatement of greenhouse gases through the implementation of policies like carbon taxes, emissions trading, measures to support specific technologies (e.g., wind, solar, carbon capture), and regional fuel and emissions standards.”

- 78/79: ungrammatical sentence

This sentence was removed, as the whole paragraph was revised.

- 128: needs a "the"

This is corrected in the revised manuscript.

- 151: Vague language: "using a different climate model" cannot "have a major impact on the effect of climate mitigation". Climate models may show large uncertainty in simulated impacts -- if that is what is meant, I think it would be best to say so in clear terms.

We split the sentence in two to improve the clarity of our statement: “By 2100, the population exposed to water stress is generally projected to increase by several hundred million under most scenarios, mainly driven by increases in water demand from a growing population. However, the use of different climate models—through statistical emulation techniques—results in contrasting estimates of the impact of climate mitigation, because of differences in regional patterns of precipitation change”.

- 166: varying

This is corrected in the revised manuscript.

- 177: cannot make sense of the ending of this sentence

This sentence was improved in the revised manuscript: “The major co-benefits of climate mitigation on air quality are consistent with existing estimates, although the actual magnitude of the co-benefits can vary substantially between studies because of differences in the scenarios and differences in the treatment of criteria pollutant emissions by different IAMs”.

- 177-179: ungrammatical and ambiguous

This sentence was improved in the revised manuscript: “Finally, the climate impacts on agricultural productivity differ from AgMIP analyses because our estimates include ozone damage and land-use change adaptation”.

- 190-192: ungrammatical

This sentence was revised: “These results also support the need to rely on probabilistic ensembles of climate simulations to determine the full range of outcomes and move into quantitative climate risk assessments. Such probabilistic impact assessment has been conducted with the IGSM for specific regions of the world.

- 217-219: ungrammatical

This sentence was revised: “Such approach could not only help quantify parametric or scenario uncertainty, but also address structural uncertainty (associated with the use of different models) by using emulators to reproduce and account for the varying behavior of different models.”

- 223-227: very long and ungrammatical sentence

This sentence was revised: “At the same time, computationally demanding process-based impact models are still required to assess the climate impacts on

specific sectors, such as air quality and health. The need for state-of-the-art models is well illustrated by recent evidence of the important role of natural climate variability on regional atmospheric chemistry, further questioning the adequacy of damage functions based on global mean temperature.”

- 231: wide range

This is corrected in the revised manuscript.

- 231: What are "responsive policy scenarios"?

We replaced “responsive” with “up-to-date”, meaning that the scenarios are responding to the latest development in climate policies.

- 183-185: This is a very bleak take on the state of economic assessment of climate impacts. There is much and very active progress on this front, and you mention some counterexamples further below in the text.

We revised the sentence: “Despite being an active area of research, there is little theory to guide the damage functions needed to directly translate change in global mean temperature to impacts on Gross Domestic Product (GDP), and in many cases arbitrary functional forms and corresponding parameter values are chosen.”

Reviewer #3 (Remarks to the Author):

The authors have done a nice job addressing my comments and the comments of the other reviewers. I have a few small, editorial comments remaining:

We thank the reviewer for his comments, which helped improve the manuscript significantly.

Line 33: Should be “it results” instead of “its results”

This is corrected in the revised manuscript.

Lines 144-149: Sentence is long and confusing, consider revising

This sentence was split in two and should be clearer: “China and India, two countries that currently experience severely polluted ambient air (with annual mean concentrations of PM_{2.5} greater than air quality standards over major areas), would see increased pollution by 2100 under the Paris Forever scenario, with PM_{2.5} concentrations doubling in many regions. However, these countries would experience significant co-benefits of imposing a carbon tax under the 2C scenario with reductions in co-emitted air pollutants including PM_{2.5}.”

Line 223: “computationally demanding”?

This is corrected in the revised manuscript.

REVIEWERS' COMMENTS:
Only comments to the editor